# Annealed Low Energy States in Frustrated Large Square Josephson Junction Arrays

**Martijn Lankhorst [1],\***  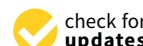**, Alexander Brinkman [1], Hans Hilgenkamp [1], Nicola Poccia [2] and Alexander Golubov [1,3]**

[1] MESA+ Institute for Nanotechnology, University of Twente, 7500 AE Enschede, The Netherlands; a.brinkman@utwente.nl (A.B.); j.w.m.hilgenkamp@utwente.nl (H.H.); a.a.golubov@utwente.nl (A.G.)
[2] Department of Physics, Harvard University, Cambridge, MA 02138, USA; npoccia@g.harvard.edu
[3] Moscow Institute of Physics and Technology, Dolgoprudny, 141700 Moscow Region, Russia
\* Correspondence: m.lankhorst@utwente.nl

**Abstract:** Numerical simulations were done to find low energy states in frustrated large square Josephson Junction arrays in a perpendicular magnetic field using simulated annealing on the coupled RSJ model. These simulations were made possible by a new algorithm suitable for parallel gpu computing and reduced complexity. Free boundary conditions were used so that values of the frustration factor $f$ that are incommensurate with the array size are permitted. The resulting energy as a function of $f$ is continuous with logarithmic discontinuities in the derivative $dE/df$ at rational frustration factors $f = p/q$ with small $q$, substantiating the mathematical proof that this curve is continuous and further showing that the staircase state hypothesis is incorrect. The solution shows qualitative similarities with the lowest energy branch of the Hofstadter butterfly, which is a closely related problem. Furthermore, it is found that at the edge of an array there are either extra vortices or missing vortices depending the frustration factor, and the width of this region is independent of the array size.

**Keywords:** Josephson junction array; superconductivity; magnetism

## 1. Introduction

The ground state configurations of infinite square Josephson Junction arrays in a uniform perpendicular magnetic field have been studied in great detail in the past [1–12], but the problem is not fully solved. It was found that such a system exhibits Josephson vortices with logarithmic long range interactions [13,14]. It is known that the mean number of vortices per unit cell is equal to the frustration factor $f = \Phi/\Phi_0$ where $\Phi$ is the magnetic flux threading a unit cell. Furthermore, at rational values of the frustration factor $f = p/q$ it was suspected that ground states have vortex configurations which are periodic with a unit cell of size $q$ by $q$ [1], but it was later shown that for some fractions a lower energy state can be found with a tile of size $2q$ by $2q$ [4,11]. For small fractions, the elementary tiles are known [11]. Many fractions between $f = 1/3$ and $f = 1/2$ have diagonal striped vortex patterns called staircase states, which led Halsey to hypothesize that this would be the case for all values [2]. The energy corresponding to these states is $\frac{E}{E_J} = 1 - \frac{1}{q}\csc(\frac{\pi}{2q})$, which only depends on $q$, so it is not a continuous function. It has the property that it is constant for irrational values of $f$, where it is equal to $1 - \frac{2}{\pi}$. However, a counterexample with lower energy was first found in [3] and later several more counterexamples were found [11]. Furthermore, it was proven mathematically that the energy as a function of frustration factor must be continuous [10], which contradicts the staircase state hypothesis.

For frustration factors with large denominators only a few examples are known. This requires a large elementary tile, and the number of possible vortex configuration grows as $\binom{q^2}{pq}$ so a brute force

search is out of the question. Previously, Monte Carlo simulations were used on arrays with periodic boundary conditions. It was done for multiple array sizes, because for a particular array size $N$ by $N$ only frustration factors that are commensurate with $N$ are allowed [11]. This commensurability requirement results in gaps around frustration factors with small $q$, which in turn makes it hard to numerically check the statement that the $E(f)$ must be continuous. Furthermore, the behavior at frustration factors close to these small fractions is not fully understood except around $f = 0$ where the problem reduces to that of a Coulomb gas [15] and the vortex configurations are strained triangular lattices superimposed on a square lattice [11]. Knowledge of these ground state configurations can help understanding the recent experiments in the current driven regime [16,17].

In this work an alternative approach is taken. Firstly, simulations were done on a large array with free boundary conditions, so one does not have to obey the commensurability requirement. Secondly, simulated annealing is done on time dependent simulations. The sparsity pattern of the linear system that has to be solved at each timestep allows it to be solved with the Fast Poisson Solver technique, which has a complexity of $\mathcal{O}(N^2 \log(N))$ per timestep for an $N$ by $N$ array, as opposed to a complexity of $\mathcal{O}(N^3)$ for a Newton-method type algorithm reported in [8,9]. Furthermore, this algorithm is suited for gpu computing which significantly reduces computation time. This better scaling allows the study of larger array sizes, although it is possible that previous methods do require less timesteps to converge. With this method the continuity and the behavior close to small fractions can be studied.

The resulting energy as a function of frustration factor shows similarity to the lowest energy branch of the Hofstadter butterfly [18]. A comparison with the results from superconducting networks [19–21] is performed.

Free boundary conditions cause edge effects that would be not be present with periodic boundary conditions. The effect on the energy as a function of magnetic field investigated.

This work is structured as follows. First, the numerical model is described. Then, the $E(f)$ curve that was obtained and its implications are discussed. After that the vortex configurations and the edge effects are discussed, and finally a comparison is made with the lowest branch of the Hofstadter butterfly.

## 2. Methods

The problem can be stated in terms of the frustrated XY model reported in [1], see Equations (1)–(3). This models an N by N square array of overdamped Josephson junctions. $I_c$ and $R_n$ are assumed to be the same for all junctions. Furthermore, screening is neglected.

$$\mathcal{H} = \sum_{<ij>} E_J(1 - \cos(\phi_{ij})) \tag{1}$$

$$\phi_{ij} = \text{pv}(\theta_j - \theta_i - A_{ij}) \tag{2}$$

$$\sum_{\text{loop}} \phi_{ij} = 2\pi\left(\frac{\Phi_{\text{enc}}}{\Phi_0} - \sum n_{\text{enc}}\right) \tag{3}$$

The ground states can be found by minimizing the Hamiltonian in Equation (1) in terms of the superconducting phases $\theta_i$ at each node. The Hamiltonian is defined in terms of the gauge invariant phase difference $\phi$, which is defined in terms of the superconducting phases $\theta$ on the nodes in Equation (2). Its principal value (pv) is taken in the interval $[-\pi, \pi)$. The quantity $A_{ij}$ is defined as $\frac{2\pi}{\Phi_0} \int_i^j \vec{A} \cdot d\vec{r}$. Here $\vec{A}$ is the magnetic vector potential. A perpendicular magnetic field is applied and a gauge is chosen such that $A_{ij}$ is zero for vertical junctions and $2\pi f y$ for horizontal junctions where $f = \frac{\Phi}{\Phi_0}$; i.e., the frustration factor defined as the magnetic flux treading a unit cell divided by the flux quantum. Furthermore, $y$ is the vertical coordinate of the junction. The Josephson Energy $E_J$ is equal to $\Phi_0 I_c$.

Equation (3) is the quantization rule. The sum is taken along any closed path through the array, $\Phi_{enc}$ is the magnetic flux through that path and $n_{enc}$ is the number of the Josephson vortices within the path. On a local minimum of $H$ in terms of $\theta$, the winding rules are automatically satisfied and the vortex configuration can be deduced from them.

For a given frustration, multiple local minima exist, and each local minimum corresponds to a configuration of Josephson vortices. The amount of local minima to the Hamiltonian for large systems grows strongly with the array size and becomes too large for a brute-force search. To find low energy states, simulated annealing is used on time dependent simulations. This is done with the RSJ model.

In the RSJ model [22,23], the current $I_{ij}$ through a junction between node $i$ and node $j$ is given by:

$$I_{ij} = I_c \sin\left(\theta_j - \theta_i - A_{ij}\right) + \frac{\hbar}{2eR_n}\frac{d(\theta_j - \theta_i)}{dt} + \eta_{ij} \tag{4}$$

Here $\theta_i$ is the superconducting phase of node $i$. The current is split in three parts; the supercurrent through the Josephson element, the current through the parallel resistor and the Johnson noise generated in the parallel resistor. The Johnson noise has a correlation defined as $\langle \eta_{ij}(t)\eta_{ij}(t')\rangle = \frac{2T}{R_n}\delta(t - t')$.

The system of equations is obtained by enforcing Kirchhoff's rules on each node. This results in a stochastic system of first order non-linear ordinary differential equations in time. This is solved numerically by applying a forward finite difference scheme in time. The stationary points of this system correspond to the minima of the Hamiltonian in Equations (1)–(3). The time discretization is worked out in Appendix A. It is written in dimensionless quantities $T' = 2\pi\frac{k_B T}{E_J}$, $\tau = (2eI_cR_n/\hbar)t$. Then, at each timestep a linear system has to be solved of size $N^2$. This is done using the fast Poisson solver technique which is described in Appendix B [24]. Furthermore, the computation was done on a GPU for which this algorithm is well suited. The code was written in CUDA.

Simulations were done with $N = 50$, $N = 100$ and $N = 512$. The time step was set to $\Delta\tau = 0.5$. The annealing procedure was done for each value of $f$ separately. For $N = 100$, a run at a given $f$ contains $10^5$ timesteps where $T'$ is kept constant at 0.1 for the first 20,000 steps and then linearly quenched to 0.

First, at a specific value of $f$, 100 runs are done in parallel with random initial conditions. With random is meant that the phase at each island is drawn from a uniform distribution between 0 and $2\pi$ at $t = 0$. All runs end up in a state with a particular vortex configuration and energy. The second iteration again consists of 100 runs, but this time the phases at the final timestep of the first iteration were used as initial conditions. Now, out of the combined states at the end of the first iteration and at the end of the second iteration, the 100 best states with lowest energy were chosen as initial conditions for the third iteration. In total 50 iterations were done where the lowest energy states that were found up to that point are always taken as initial conditions for the subsequent iteration. This process was done for frustration factors between 0 and 0.5 in steps of 0.001.

## 3. Results

The obtained lowest energy states as a function of frustration factor are shown in Figure 1. Simulations were only done for $f \in [0, 1/2]$ and the data is reflected with respect to $f = 1/2$, using the symmetry $f' \to 1 - f$, $n' \to 1 - n$, $\forall i\ \theta'_i \to -\theta_i$. The lowest energy states form a smooth curve except at rational values of the frustration factor where the derivative $dE/df$ shows discontinuities in line with fractal behavior. The fractions at which the jumps in $dE/df$ exceed the noise level are highlighted. The derivative was computed numerically with a central difference scheme.

The curve appears to be continuous despite the fact that all points are independent problems starting from random initial conditions, confirming the continuity predicted by [10]. This contradicts with the staircase state hypothesis. Furthermore, for a significant portion of the $f$ range between 1/3 and 1/2 with large $q$, staircase states have significantly larger energy than the lowest energy states found with annealing. For these frustration factors the staircase states are likely not groundstates. Only slightly above and below $f = 2/5$ the energy for staircase states is lower than the lowest energy

states found with annealing. The energy of the annealed states does still go down with longer annealing time and the energy difference is small enough that it is possible that for long enough annealing time the entire curve will be below or at the staircase state energy.

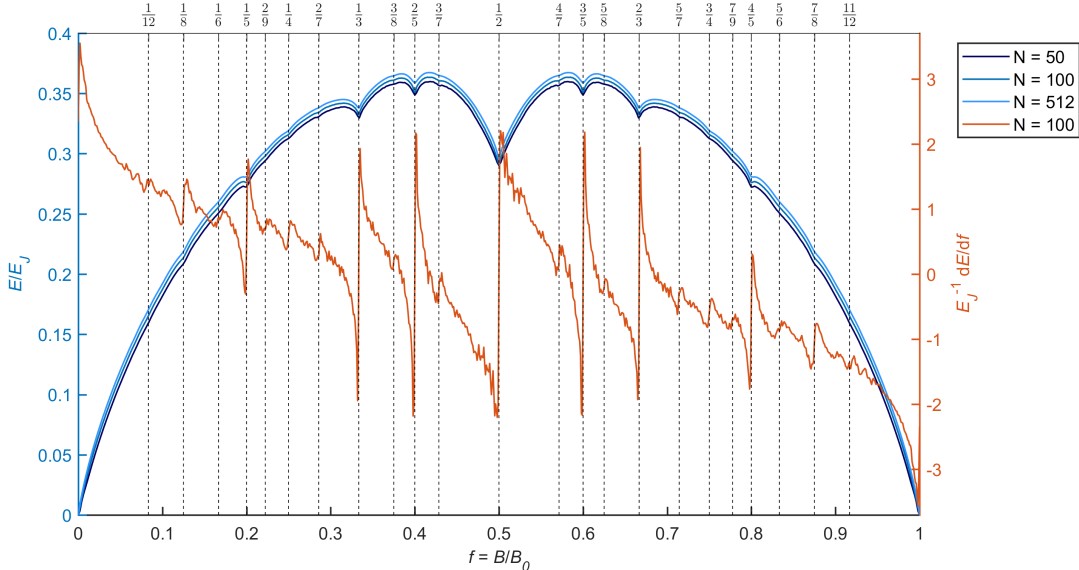

**Figure 1.** Left: Lowest energy states found as a function of frustration factor. It appears to be a continuous curve. Right: $dE/df$ showing discontinuities at rational values of the frustration factor. These discontinuities have logarithmic character.

The lowest energy states depend on the size of the array. In Figure 2 the energy is plotted as a function of array size for the small fractions $f = 1/3, 2/5, 1/2$. Array sizes are taken to be integer multiples of $q$ and repetitions of the elementary tile are used as vortex patterns. For all these fractions the ground state energy converges to the limiting value as $E(N, f)/E(\infty, f) = 1 - \lambda(f)/N$ for large $N$. $\lambda(1/2) = 0.45$, $\lambda(1/3) = 0.56$, $\lambda(2/5) = 0.57$ were found. The limiting value $E(\infty, f)$ is known for these small fractions (it is $\frac{1}{3}$, $\frac{4-\sqrt{5}}{5}$ and $\frac{2-\sqrt{2}}{2}$ respectively). It was found that the junctions at the edge of the array have a lower Josephson Energy than junction in the center of the array. The width of this region is characterized by $\lambda$, which is independent on array size but dependent on $f$.

In Figure 3 the vortex patterns on the center 21 by 21 tiles for some fractions are shown. Similar patterns have been observed in magnetic probe measurements on superconducting networks [25,26]. The patterns look as follows. For fractions $f = 1/3, 2/5, 1/2$ the vortex pattern that was found is equal to the tiled patterns corresponding to the ground state of infinitely large arrays. For other small fractions, $f = 1/7, 1/6, 1/4$ for example, this tiling is not retrieved but rather grains of the elementary tiling are present separated by grain boundaries. The number of grain boundaries gets smaller for longer annealing time, and it is plausible that these grain boundaries will completely disappear for sufficiently long annealing time.

For frustration factors close to small fractions $f = 0$ and $f = 1/2$ the vortex pattern looks to be a superposition of excitations on top of the vortex pattern corresponding to that fraction, which was also found in [7]. It is known that in the regime $|f| << 1$ the vortex excitations are spaced far apart and can be treated as independent vortices which have logarithmic interaction and obey $dE/df \propto \text{sgn}(f) \log |f|$ [27]. The derivative of the lowest energy curve has logarithmic singularities not just at $f = 0$, but also at small fractions $f_c$ where the graph obeys $dE/df \to \text{sgn}(f - f_c) \log |f - f_c|$. This suggests that for values of $f$ close to these small fractions, the vortex pattern is a superposition of base pattern plus a superlattice of defects.

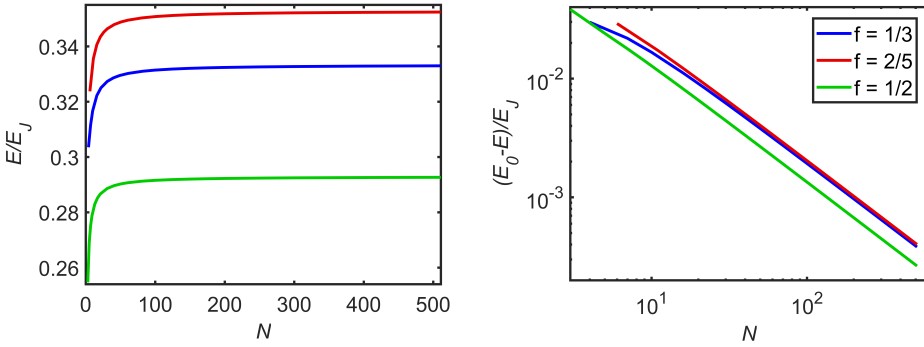

**Figure 2.** Energy as a function of array size for $f = 1/3$, $f = 2/5$ and $f = 1/2$. The energy for an infinite array, $E_\infty$ is known and is $\frac{1}{3}$, $\frac{4-\sqrt{5}}{5}$ and $\frac{2-\sqrt{2}}{2}$ respectively. and the data scales as $E(N)/E_\infty = 1 - \lambda(f)/N$. The values $\lambda(1/3) = 0.56$, $\lambda(2/5) = 0.57$ and $\lambda(1/2) = 0.45$ were obtained from fitting the data.

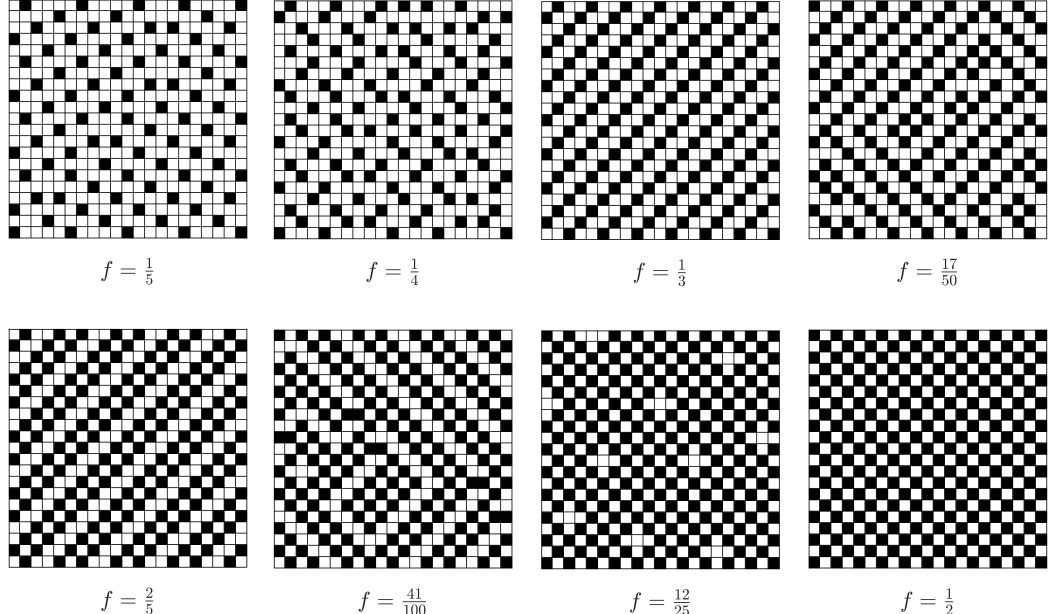

**Figure 3.** Centre 21 by 21 tiles of the vortex patterns corresponding to the lowest energy states found in a 100 by 100 array for some selected fractions.

In Figure 4, the full vortex configuration for $f = 1/5$ and $N = 100$ is shown. It is clear that vortices are missing at the edge. For an infinite array, the mean number of vortices is exactly equal to the frustration factor, but for a finite array it is allowed to deviate slightly. In Figure 5 this difference times the array size is plotted as a function of frustration factor and array size. It appears that this follows a universal curve and the difference between the mean number of vortices $\bar{n}$ and $f$ scales as $1/N$ just like the energy. This again implies that there is a field dependent depletion- or accumulation-width at the edge of the array, although this width is not the same as the energy relaxation width.

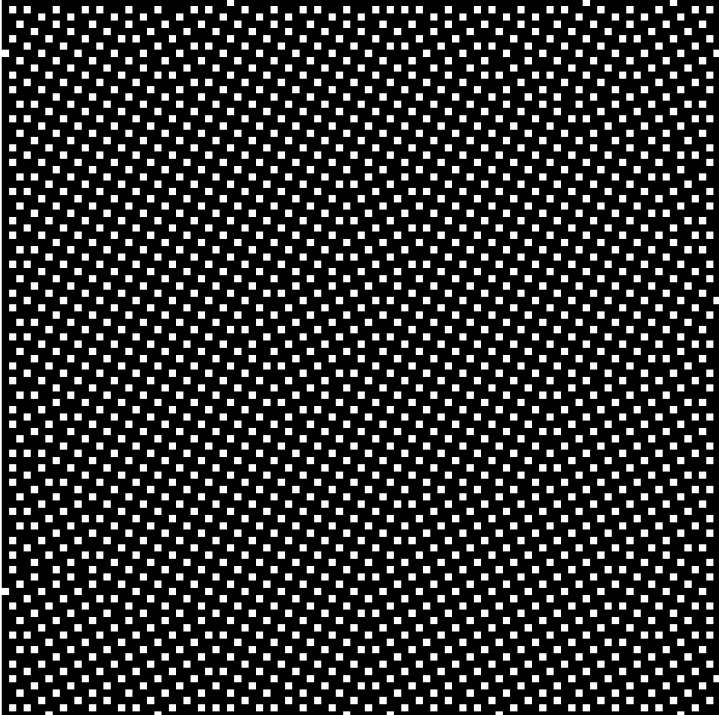

**Figure 4.** Full vortex configuration found at $f = 1/5$ and $N = 100$. Grain boundaries are observed, but the grains grow as the annealing time is increased so it is likely that the grain boundaries will eventually disappear. Also missing vortices can be observed at the edge of the array due to the free boundary conditions.

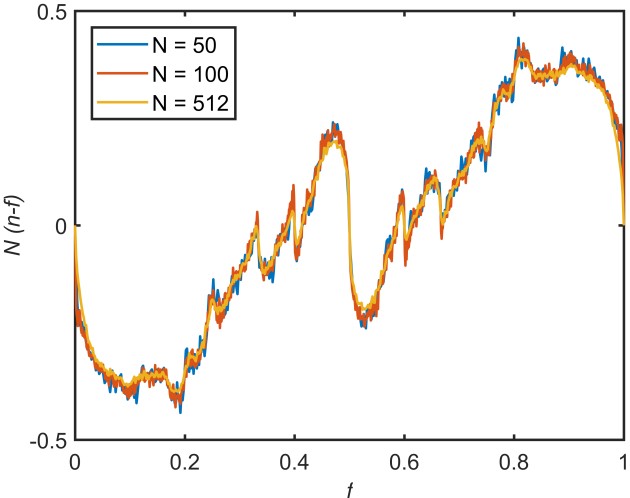

**Figure 5.** Difference between mean number of vortices and the frustration factor times the array size, $N(\bar{n} - f)$. It converges to a universal curve, so $(\bar{n} - f)$ converges to zero as $1/N$. The extra or missing vortices are found in a region of constant width at the edge of the array, and this width is dependent on the frustration factor.

In Figure 6 the lowest energy states of the frustrated XY model are compared to the lowest energy branch of the Hofstadter butterfly which is proportional to the $T_c$ for superconducting networks as predicted by the linearized GL network equations [18–21]. The general shape of the curve is very similar and both show fractal behavior with apparent singularities at rational values of $f$. However, these singularities seem of different type. In the frustrated XY model these singularities appear logarithmic, while in the Hofstadter butterfly the singularity is approached linearly from both sides. The similarity of these curves suggests there might be a connection between the ground state energy of JJAs and the $T_c$ for superconducting networks.

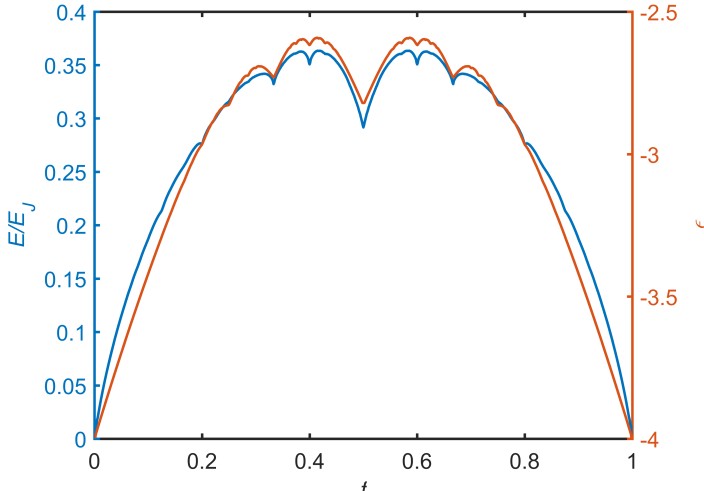

**Figure 6.** Left: Lowest energy states found for a 100 by 100 square Josephson junction array using the frustrated XY model. Right: Lowest energy branch of the Hofstadter butterfly, which is proportional to the $T_c$ of a superconducting square network.

## 4. Conclusions

Low energy states of large finite square Josephson junction arrays have been computed with an annealing algorithm. The resulting curve looks continuous which is in agreement with a proof of continuity. It gives further evidence that the staircase state hypothesis is incorrect. The curve shows logarithmic discontinuities in the derivative $dE/df$ at rational frustration factors $f = p/q$ with small $q$. This is consistent with the picture of independent vortex excitations on top of the base vortex pattern when $f$ is close to a simple fraction. These ground state vortex patterns of finite arrays seem the same as the ground state for infinite arrays, except for a thin region at the edge of the array where vortices are accumulated or depleted, depending on the frustration factor. However, the width of this region is array size independent.

**Author Contributions:** M.L. Conceived and designed the experiment, performed the simulations and analyzed the data. A.B., H.H., N.P. and A.G. helped interpret the results. M.L., N.P. and A.G. wrote the paper.

**Acknowledgments:** This work was supported by the Dutch FOM and NWO foundations.

**Conflicts of Interest:** The authors declare no conflict of interest.

## Appendix A

The goal of this section is to precisely state the system of differential equations describing a square array of Josephson junctions. In the next section an algorithm to solve this system is described.

The current on a single junction after applying a finite difference scheme in time is given by Equation (A1).

$$J_{ij} = \sin\left(\theta_j^n - \theta_i^n - A_{ij}\right) + \frac{(\theta_j^{n+1} - \theta_i^{n+1}) - (\theta_j^n - \theta_i^n)}{\Delta\tau} + 2\sqrt{\frac{T'}{\Delta\tau}} G_{ij}^n \tag{A1}$$

Here, $J_{ij}$ is the current from node $i$ to node $j$, $\theta_i$ is the superconducting phase at node $i$, $A_{ij} = \frac{2\pi}{\Phi_0}\int_i^j \vec{A}\cdot d\vec{r}$ where $\vec{A}$ is the magnetic vector potential, $T' = 2\pi\frac{k_B T}{E_J}$, $\tau = (2eI_cR_n/\hbar)t$, $n$ is the timestep number and $G_{ij}^n$ are Gaussian random numbers with $\mu = 0$ and $\sigma = 1$. The full system is obtained by applying Kirchhoff's rules on each lattice site. The system is solved iteratively and at each timestep a linear system of equations has to be solved. To write down this system it is convenient to number the nodes systematically and to define the gauge invariant phase differences $h_{ij} = \theta_{ij+1} - \theta_{ij} + 2\pi fia$ for horizontal junctions and $v_{ij} = \theta_{i+1j} - \theta_{ij}$ for vertical junctions, see Figure A1. Here $a$ is the lattice constant. Furthermore, the phases of a row of nodes are collected into a vector: $\vec{\theta}_i = [\theta_{i1}, \theta_{i2}, \ldots, \theta_{iN}]^T$. Similarly, $\vec{h}_i = [h_{i1}, h_{i2}, \ldots, h_{iN-1}]^T$ and $\vec{v}_i = [v_{i1}, v_{i2}, \ldots, v_{iN}]^T$. The full system is then a linear system of size $N^2$, see Equation (A2).

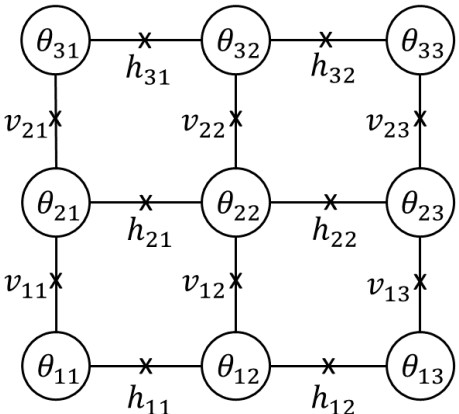

**Figure A1.** Numbering convention in a 3 by 3 array.

$$\begin{bmatrix} A-I & -I & & & & \\ -I & A & -I & & & \\ & -I & A & \ddots & & \\ & & \ddots & \ddots & -I & \\ & & & -I & A & -I \\ & & & & -I & A-I \end{bmatrix} \begin{bmatrix} \vec{\theta}_1^{n+1} \\ \vec{\theta}_2^{n+1} \\ \vec{\theta}_3^{n+1} \\ \vdots \\ \vec{\theta}_{N-1}^{n+1} \\ \vec{\theta}_N^{n+1} \end{bmatrix} = \begin{bmatrix} \vec{r}_1^n \\ \vec{r}_2^n \\ \vec{r}_3^n \\ \vdots \\ \vec{r}_{N-1}^n \\ \vec{r}_N^n \end{bmatrix} \tag{A2}$$

The right hand sides $\vec{r}_i$ are defined as:

$$\vec{r}_i^n = \begin{cases} (A-I)\vec{\theta}_1^n - \vec{\theta}_2^n - \Delta\tau(D\vec{J}_1^{h,n} - \vec{J}_1^{v,n}) & i = 1 \\ A\vec{\theta}_i^n - \vec{\theta}_{i-1}^n - \vec{\theta}_{i+1}^n - \Delta\tau(D\vec{J}_i^{h,n} + \vec{J}_{i-1}^{v,n} - \vec{J}_i^{v,n}) & 1 < i < N \\ (A-I)\vec{\theta}_N^n - \vec{\theta}_{N-1}^n - \Delta\tau(D\vec{J}_N^{h,n} + \vec{J}_{N-1}^{v,n}) & i = N \end{cases} \tag{A3}$$

With $A$ and $B$ defined as:

$$
A = \begin{bmatrix}
3 & -1 & & & & & \\
-1 & 4 & -1 & & & & \\
& -1 & 4 & \ddots & & & \\
& & \ddots & \ddots & -1 & & \\
& & & -1 & 4 & -1 & \\
& & & & -1 & 3
\end{bmatrix}
\tag{A4}
$$

$$
D = \begin{bmatrix}
1 & & & & & \\
-1 & 1 & & & & \\
& -1 & \ddots & & & \\
& & \ddots & 1 & & \\
& & & -1 & 1 & \\
& & & & -1
\end{bmatrix}
\tag{A5}
$$

And the normalized current *J* is defined as the sum of the supercurrent and fluctuations:

$$
\vec{J}_i^{h,n} = \sin \vec{h}_i^n + 2\sqrt{\frac{T'}{\Delta\tau}} \vec{G}_i^{h,n}
\tag{A6}
$$

$$
\vec{J}_i^{v,n} = \sin \vec{v}_i^n + 2\sqrt{\frac{T'}{\Delta\tau}} \vec{G}_i^{v,n}
\tag{A7}
$$

**Appendix B**

Calculating the superconducting phases at subsequent timesteps is done by solving a banded linear system of size $N^2$. This has complexity $\mathcal{O}(N^3)$ when done with banded solver, but it can be improved to a complexity of $\mathcal{O}(N^2 \log N)$ by using the Fast Poisson Solver technique [24].

The linear system to be solved is given by Equation (A2). The sparsity pattern of the matrix allows the linear system to be solved with a fast Poisson solver. It involves three steps. First, the matrix *A* is diagonalized as follows:

$$
A = Q\Lambda Q^T
\tag{A8}
$$

With $I = QQ^T = Q^TQ$, because *A* is symmetric. Furthermore, $\Lambda$ is the diagonal eigenvalue matrix with $\Lambda_{ii} = \lambda_i$. The matrices *Q* and $\Lambda$ have analytic expressions, see Equations (A9)–(A11).

$$
\lambda_i = 4 - 2\cos(\pi(i-1)/N)
\tag{A9}
$$

$$
\vec{x} = [1/2, 3/2, \dots, N - 1/2]^T \times \pi/N
\tag{A10}
$$

$$
Q = [\vec{1}, 2\cos(\vec{x}), 2\cos(2\vec{x}), \dots, \cos((N-1)\vec{x})]\sqrt{N}
\tag{A11}
$$

Step 2 is done by substituting *A* in Equation (A2) by Equation (A8) and then left multiplying by $Q^T$ and right multiplying by *Q*. This leads to Equation (A14):

$$
\vec{\theta}_i' = Q^T \vec{\theta}_i
\tag{A12}
$$

$$
\vec{r}_i' = Q^T \vec{r}_i
\tag{A13}
$$

$$
\begin{bmatrix}
\Lambda - I & -I & & & & \\
-I & \Lambda & -I & & & \\
& -I & \Lambda & \ddots & & \\
& & \ddots & \ddots & -I & \\
& & & -I & \Lambda & -I \\
& & & & -I & \Lambda - I
\end{bmatrix}
\begin{bmatrix}
\vec{\theta}_1' \\
\vec{\theta}_2' \\
\vec{\theta}_3' \\
\vdots \\
\vec{\theta}_{N-1}' \\
\vec{\theta}_N'
\end{bmatrix}
=
\begin{bmatrix}
\vec{r}_1' \\
\vec{r}_2' \\
\vec{r}_3' \\
\vdots \\
\vec{r}_{N-1}' \\
\vec{r}_N'
\end{bmatrix}
\tag{A14}
$$

The last step is splitting this system into $N$ independent linear tridiagonal systems of size $N$ (Equation (A15)). One can obtain $\vec{\theta}_i'$ by solving the $i$-th tridiagonal system and subsequently obtain $\vec{\theta}_i$ by left multiplying with $Q$. Then $\forall i \in [1, N]$:

$$
\begin{bmatrix}
\lambda_i - 1 & -1 & & & & \\
-1 & \lambda_i & -1 & & & \\
& -1 & \lambda_i & \ddots & & \\
& & \ddots & \ddots & -1 & \\
& & & -1 & \lambda_i & -1 \\
& & & & -1 & \lambda_i - 1
\end{bmatrix}
\begin{bmatrix}
\theta_{1i}' \\
\theta_{2i}' \\
\theta_{3i}' \\
\vdots \\
\theta_{N-1i}' \\
\theta_{Ni}'
\end{bmatrix}
=
\begin{bmatrix}
r_{1i}' \\
r_{2i}' \\
r_{3i}' \\
\vdots \\
r_{N-1i}' \\
r_{Ni}'
\end{bmatrix}
\tag{A15}
$$

A final note is that left multiplying with $Q^T$ is the same operation as applying a discrete cosine transform of the second kind (dctII). This can be solved with an fft type of solver, which has a complexity of $\mathcal{O}(N \log N)$ rather than $\mathcal{O}(N^2)$. The final algorithm requires solving $N$ tridiagonal systems of size $N$ and $2N$ dct's.

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
