# Peer review of "Annealed Low Energy States in Frustrated Large Square Josephson Junction Arrays"

_condensedmatter, doi:10.3390/condmat3020019_

Round 1

Reviewer 1 Report

This manuscript presents numerical simulations to find low energy states in frustrated large square Josephson junction arrays using simulated annealing on the coupled RSJ model. The authors employed a new algorithm suitable for gpu computing.

I believe the results are confusing therefore I cannot recommend publication of this manuscript. Since (a) only numerical results have been offered in the manuscript, (b) the resulting energy as a function of the perpendicular magnetic field looks continuous which is in disagreement with the expected staircase behavior, and (c) the curve comparison in the Figure 5 is not perfect, one can question either precision of the calculations or even whether simulated annealing methodology using the coupled RSJ model is working well to calculate low energy states in frustrated large square Josephson junction arrays. The authors need to carefully address these questions and offer solid explanations of the observed behavior or alternatively recheck their computation.

Author Response

Thanks for this referee's comments.

This referee has three main criticisms which we will address in turn.

The first criticism is that the manuscript only show numerical results. This is correct, but we believe this work still falls into the scope of the journal.

The second point is that the energy as a function of magnetic field looks continuous which contradicts with the staircase state hypothesis. We believe this is not due to erroneous data but that the staircase state hypothesis is incorrect. In fact, it is proven mathematically that the energy as a function of magnetic field is continuous (see Vallat and Beck, Phys. Rev. Lett. 68, 3096 (1992)). Furthermore, several counterexamples have been found showing non-staircase vortex configurations which are lower in energy than the states predicted by Halsey (PRB 48, 3309 (1993)).

The third point is that in figure 5 where the energy is compared to the lowest branch of the Hofstadter butterfly, there is no perfect correspondence. Again, we believe that these to curves should not be equal. In (PRB 48, 3309 (1993)) it is shown that lowest branch of the Hofstadter is the solution of a mathematically different system, and they explicitly show the solutions are different for multiple frustration factors.

Reviewer 2 Report

The paper by Lankhorst et al. studied the ground state and its 

energy in the frustrated Josephson junction array (FJJA) with 

respect to the frustration factors f = p/q through the numerical 

annealing simulations. The main finding is that the energy 

curve is continuous as a function of f. The authors also 

discussed the boundary effect due to the finite size system 

of the simulations. 

In the present manuscript, I cannot find new results 

in the problem of the FJJA. 

The continuity of the ground state energy as a function of 

the filling factor f=p/q has been discussed in the paper 

Vallat and Beck, Phys. Rev. Lett. 68, 3096 (1992). 

Finite size effect of the ground state configuration has also 

been studied by some authors, especially for irrotational 

fillings, e.g., PRB 56, 95 (1997), PRB 59, 9569 (1999), 

PRB 60, 3163 (1999), PRL 85, 3484 (2000), etc. 

When the paper were resubmitted, I would like to know 

whether the paper can give any progress in the understanding 

of FJJA from the previous literatures. 

If the annealing algorithm is novel, I think that the 

paper would be suitable for publication in some journal 

of numerical physics. 

Anyway, the terminology "stimulated annealing" must be 

"simulated annealing".

Author Response

Thanks for this referee's comments.

This referee shows several references which were not cited in the manuscript concerning continuity of the ground state and the influence of finite size effects due to free boundary conditions.

The paper by Vallat and Beck, (PRL 68, 3096 (1992)) where it is mathematically proven that the ground state energy is a continuous function of E is a reference we missed, and acknowledge that this is a key reference and should be cited. We however still believe our manuscript gives new insight on the matter. Previous works that investigated the ground state energy as a function of f used periodic boundary conditions, which limits the values of f that are commensurate with the array size. The works (PRL 51, 1999 (1983)), (PRB 48, 3309 (1993)) show gaps in the spectrum due to this problem. Because we used free boundary conditions, we could take many values of f, filling the gaps. With this, continuity can be explicitly checked and it indeed confirms the mathematical proof by (PRL 68, 3096 (1992)). Furthermore, it provides new insight in the behavior close to rational values of f with small denominator. In fact it the derivative clearly shows discontinuities at these values, and the asymptote is approached logarithmically.

The other references that were mentioned investigated free boundary conditions and compared free to open boundary conditions for some values of f and several array sizes. It is described that at the edge of the array the energy density decreases exponentially. The result shown in this paper that this decay width is independent of array size is not mentioned, but perhaps not a surprising result. However, the description of missing or excess vortices at the edge of the array showing that this is a complex function of magnetic field but again independent of array size, is new and a nontrivial result.

Reviewer 3 Report

Comments are contained in the uploaded file.

Author Response

I will respond to referee 4 point by point.

1)      Nonperiodic boundary conditions were chosen because it allows for incommensurate values of f and the authors other interest in dynamic simulations, thus the algorithm was developed for this. I do believe the algorithm could be adjusted to work for periodic arrays as well.

2)      I reformulated this in the text to hopefully be clearer. 

3)      The phases on each island were assigned a uniform random number between $0$ and $2\pi$.

4)      A discrete difference was used (central difference scheme)

5)      It is not because of free boundary conditions (as this only lowers the energy), it is because the true groundstate is not yet reached and defects and grain boundaries are still present. These grain boundaries clearly shrink as the annealing time is increased, so I believe with a sufficiently long annealing time the energy will end up below or at the Harper limit for all values of $f$.

6)      I adjusted the main text to explain this point more clearly.

7)      I believe it does not depend on whether one uses periodic of free boundary conditions, but on the annealing time. The grain boundaries would go away if a sufficiently long annealing time is used. The text has been adjusted to explain this more clearly.

8)      Indeed no external current was used, this was a mistake.

9)      The mistake is corrected.

10)   I included both references.

Reviewer 4 Report

Report on manuscript 304508
"Annealed low energy states in frustrated large square Josephson Junction Arrays"

By Martijn Lankhorst, Alexander Brinkman, Hans Hilgenkamp, Nicola Poccia, and
Alexander Golubov.

The manuscript describes very interesting work on the minimum energy states of square Josephson junction arrays of different sizes, studied by an approach that, to my knowledge, has hitherto been not (or little) attempted : stimulated annealing. The results, that can be summarized as :

(1) the reproduction of know Josephson vortex arrangements for elementary (simple) frustration fractions;

(2) the realization of "polycrystalline" states in which domains of the elementary states combine to form lowest energy states for non-trivial frustration fractions;

(3) an illustration of how edge effects are important;

are illuminating and very helpful, not in the least because the chosen simulation procedure most likely comes closed to what an actual experiment would produce. Even if the found states might not correspond to absolute minima of energy, they are very close, and are likely to what be commonly found due to the influence of non-zero temperature. The results are put into perspective, with an interesting comparison to the well-known features of the infinite JJ array and to the Hofstadter butterfly.

Beyond this, the work provides interesting perspectives on possible simulations of Josephson vortex (phase) dynamics in square JJ arrays.

The authors may wish to carefully proofread the manuscripts, as there are minor glitches here and there (reference to Figure 3 instead of Figure 4 on line 116, lack of definition of the subscript "enc" (probably, "enclosed") and the abbreviation "pv" (principal value) in Equation 1, several typos).

Otherwise, this fine work should be published as is.

Author Response

We have carefully proofread the paper and fixed the glitches that were mentioned.

Round 2

Reviewer 2 Report

I read the authors reply and revised manuscript. 

The authors replied properly on my comments and revised the manuscript. 

Thus, I have no objections to the acceptance of this paper.